# Towards Standardised AI Documentation: A Knowledge Graph Approach for Model and Data Cards

Andy Donald[1,*], Talha Iqbal[1], Ihsan Ullah[1], Huan Chen[1], Edward Curry[1],
Apostolos Galanopoulos[2], Emir Munoz[2] and Sagar Saxena[2]

[1]*Insight Centre for Data Analytics, University of Galway, IDA Business Park, Daingean, H91 AEX4, Galway, Ireland*
[2]*Genesys, Bonham Quay, Dock Road, The Docks, Galway, H91 AX8R, Ireland*

## Abstract

The proliferation of AI model cards and data cards has advanced transparency in machine learning, yet these documentation artefacts remain largely free-form and noncomparable. We present a novel framework that transforms unstructured AI documentation into a unified semantic knowledge graph (KG) using RDF (Resource Description Framework). Our approach defines an ontology encompassing key classes: Model and Dataset, and maps card fields into triples, enabling programmatic querying and reasoning. We augment the KG with an automated bias detection module that identifies content bias and represents it as structured *Bias Finding* nodes linked to the relevant entities. To facilitate comparison across Model and Dataset cards, we compute graph similarity metrics - both Graph Edit Distance (GED) and Entity Comparison in graph analysis—allowing clustering of similar models or datasets and highlighting structural documentation gaps. An airflow-orchestrated ingestion pipeline parses Markdown/JSON cards, converts them to RDF, enriches them with bias analyses and structural similarity metrics, and loads them into a GraphDB triplestore with SPARQL endpoints and a user-friendly dashboard. We demonstrate the utility of our system through case studies: clustering and gap analysis of heterogeneous, natural language processing (NLP) based models and data cards, and tracing dataset bias through downstream models. Our quantitative evaluation, conducted on 10,000 HuggingFace-based models and data cards, reveals distinct domain-based clusters, achieves approximately 85% accuracy in bias detection, and provides valuable insights into the interconnections among widely used resources. This work lays the foundation for standardised, machine-readable AI documentation, fostering transparent comparison, auditability, and governance of AI systems.

## Keywords

Knowledge Graphs, AI Documentation, Model Cards, Data Cards, RDF, Ontologies, Bias Detection, Graph Similarity, Semantic Web, Responsible AI

## 1. Introduction

Transparency and accountability are critical in modern AI systems, prompting the development of documentation standards like model cards and data cards [1]. A model card is a document detailing an AI model's essential characteristics—such as its intended uses, performance metrics, and ethical considerations—in order to inform users about appropriate and inappropriate use cases [1]. Similarly, a data card provides structured documentation for datasets, describing aspects like motivation, composition, collection process, and known biases [2]. These documentation frameworks have gained traction as important tools for Responsible AI, with industry and academic efforts encouraging their adoption [3]. Despite their growing use, model cards and data cards are largely presented as free-form text or PDF reports, which limits their standardization and comparability. Each model or dataset's documentation

---

*Seventh International Workshop On Knowledge Graph Construction, Dubrovnik, Croatia - May 10–11, 2026*

*Corresponding author.

✉ andy.donald@universityofgalway.ie (A. Donald); talha.iqbal@universityofgalway.ie (T. Iqbal);
ihsan.ullah@universityofgalway.ie (I. Ullah); huan.chen@universityofgalway.ie (H. Chen);
edward.curry@universityofgalway.ie (E. Curry); apostolos.galanopoulos@genesys.com (A. Galanopoulos);
emir.munoz@genesys.com (E. Munoz); sagar.saxena@genesys.com (S. Saxena)
🌐 https://research.universityofgalway.ie/en/persons/andy-donald/ (A. Donald)
ⓘ 0009-0007-3307-2800 (A. Donald); 0000-0001-9505-2732 (T. Iqbal); 0000-0002-7964-5199 (I. Ullah); 0009-0007-8549-0414
(H. Chen); 0000-0001-8236-6433 (E. Curry); 0000-0002-3359-1922 (A. Galanopoulos); 0000-0002-0089-8135 (E. Munoz);
0000-0003-1651-4519 (S. Saxena)

may follow a slightly different format or emphasis, making it difficult to programmatically compare two models or aggregate insights across many projects. For example, one model card might list detailed fairness metrics by demographic, while another omits this information entirely. Important details—such as the presence of bias in a dataset or limitations of a model—could be described in prose, making automated retrieval or analysis challenging. In short, current AI documentation lacks a structured, machine-readable representation. This absence of standard structure hinders efforts to systematically search, query, or compare AI models and datasets on key attributes, and it impedes meta-analyses of documentation practices themselves [4]. In this paper, we propose to address these challenges by constructing a semantic knowledge graph (KG) for AI model cards and data cards. Our approach transforms the unstructured content of these documents into an interconnected, queryable graph of facts. By using a semantic framework (specifically an RDF KG) we can represent the diverse information in model/data cards (e.g., model details, evaluation results, dataset characteristics) as nodes and relationships with well-defined meaning. This structured representation enables direct comparisons and aggregations that were previously infeasible. For instance, one can query the graph to find all models trained on a certain dataset, or to enable automated inferential reasoning to trace bias propagation, such as identifying models that potentially inherit a known representational harm from a linked, biased upstream dataset. A key aspect of our approach is the integration of automated bias detection into the graph. We incorporate analytical modules that assess datasets and models for common biases (such as representation skew or performance disparities) and record these findings within the KG. By linking bias findings to the corresponding dataset or model nodes, we enrich the documentation with machine-generated insights about fairness. This not only adds valuable information that might be missing from human-written cards, but also enables reasoning over bias propagation (e.g., tracing how a dataset's imbalance might influence multiple models). Furthermore, we introduce techniques for measuring structural similarity among documentation graphs. Treating each model or dataset's documentation as a subgraph, we can apply graph similarity metrics to quantify how alike two documentation profiles are, primarily focusing on structural similarity - that is, the presence or absence of key documentation components and how they are organized within the graph. This includes comparing the types of nodes (e.g., evaluation metrics, usage guidelines, bias findings), their relationships, and the completeness of expected documentation patterns, rather than the semantic similarity of the content or attributes themselves. This allows us to, for example, cluster models by the similarity of their documentation (which may correlate with model type or domain), or highlight outliers that deviate from standard documentation patterns. Structural comparison also helps identify documentation gaps—for instance, detecting that one model's card is missing sections that are present in most others. To realize this approach, we design and implement a pipeline for ingesting, processing, and visualizing the KG. Using tools like Apache Airflow [1] for orchestration, our pipeline ingests model cards and data cards, parses their content into a unified RDF schema, and stores the results in a graph database (such as GraphDB). We provide a SPARQL endpoint for querying the graph, enabling complex questions to be answered via semantic queries. Additionally, we explore visualization techniques to browse and inspect the graph, making it easier for stakeholders to interpret relationships and findings. In summary, the contributions of this work are:

- **RDF-based KG Schema:** We develop an ontology and RDF schema to represent the contents of model cards and data cards in a structured, unified way. Key classes (e.g., Dataset, Model) and properties are defined to capture documentation fields as semantic triples.
- **Integrated Bias Detection:** We integrate automated bias detection methods into the documentation pipeline. The system identifies issues such as dataset imbalance, model performance disparities, and other biases, and encodes these findings in the KG for each corresponding dataset or model entity.
- **Structural Similarity Analysis:** We propose methods to compare and quantify the structural and content similarity between documentation graphs. Techniques such as Graph Edit Distance

---

[1]https://airflow.apache.org

(GED) [5] are leveraged to enable comparison of models and datasets, supporting use cases like model benchmarking and selection by documentation features.

- **Entity Comparison:** Beyond whole-graph metrics, we implement entity-level comparisons that focus on specific nodes (e.g. two model or dataset entities) by extracting each node's immediate neighborhood and property set. Concretely, we retrieve the set of outgoing predicates and connected nodes for each target entity (datasets used, evaluation results, consideration nodes, etc.) and compute a Jaccard or overlap score on those sets to quantify how many properties they share.
- **Querying, Visualization, and Reasoning Framework:** We implement a pipeline with a graph database backend, supporting SPARQL queries and interactive exploration of the knowledge graph. This framework allows users to query documentation at scale (e.g., find models or datasets with certain properties or biases), visualize the connections (such as model–dataset–metric linkages), and perform reasoning (like inferring potential risks from connected bias indicators).

The remainder of this paper is structured as follows. Section 2 reviews related work on model/data cards, KGs for AI metadata, bias detection in AI systems, and graph-based similarity techniques. Section 3 details our methodology, including the RDF schema design, bias detection module, structural similarity approach, and pipeline implementation. Section 4 presents use cases and an evaluation of the system, demonstrating examples of comparing publicly available data and model cards via HuggingFace [2], as well as quantitative performance metrics. Section 5 discusses the broader implications of this work, its limitations, and possible future directions. Finally, Section 6 concludes the paper.

This work intersects several areas of research. We briefly review: (a) the development of model cards and data cards for AI documentation; (b) KGs and ontologies for representing AI model information and supporting explainability; (c) approaches to bias detection in datasets and models; (d) methods for measuring structural similarity in graphs; and (e) entity comparison techniques in graph analysis.

To support the reproducibility of this work and facilitate further research in standardized AI documentation, we have made the source code, RDF mapping scripts, and a sample of the processed dataset publicly available [3].

## 2. Background and Related Work

### 2.1. Model Cards and Data Cards

Model cards were first proposed by Mitchell et al. (2019) [1] as a standardized approach to document machine learning (ML) models, providing information on model provenance, intended use cases, performance evaluation (especially on different subpopulations), and ethical considerations. Since their introduction, model cards have been increasingly adopted in both research and industry settings as a means to improve transparency. For instance, popular ML platforms and model hubs now often encourage or require model card documentation for contributed models [3]. In parallel, analogous efforts for datasets have emerged. Datasheets for Datasets [2] laid the groundwork for dataset documentation by suggesting a questionnaire-style template covering dataset motivation, composition, collection process, recommended uses, and so on. Building on this, Data Cards were later introduced as a more concise, user-friendly format for dataset documentation [6], and have been promoted by tools like the Google Data Cards Playbook [7]. While these initiatives have established the importance of documenting models and data, they are largely informal and narrative-driven. Our studies on our HuggingFace dataset have found that the content and quality of model cards can vary widely between authors and organizations. For example, as seen in Table 1, analysis of the dataset of 5,000 model cards revealed that many omit key details such as bias evaluations or limitations, despite recommendations to include them. Similarly, our examination of 5,000 dataset data cards from HuggingFace often ranges from highly detailed to extremely sparse, with no enforced schema. IBM's AI FactSheets [8] represent an

---

[2]https://huggingface.co/
[3]https://github.com/andy-donald-insight/KGC-2026

attempt to introduce more structure by providing a checklist of facts about AI services, but these too are ultimately presented as documents. The lack of a formal schema or machine-readable format means that comparing two model cards or querying a collection of cards for specific information (e.g., "find all models evaluated on WikiData") is cumbersome at best. This gap in standardization motivates our work: we seek to retain the richness of model/data cards while converting them into a structured knowledge representation that enables easy comparison and retrieval.

| Card Type | Aspect | Count | % of Total |
|-----------|--------|-------|------------|
| Model | Complete | 372 | 7.4 % |
| Model | Bias Evaluation | 79 | 1.6 % |
| Model | Limitations | 298 | 6.0 % |
| Data | Complete | 140 | 2.8 % |
| Data | Bias Evaluation | 53 | 1.1 % |
| Data | Limitations | 52 | 1.0 % |

**Table 1**
HuggingFace model and data cards gaps

## 2.2. Knowledge Graphs for AI Documentation and Explainability

KGs have become a powerful paradigm for representing complex information in artificial intelligence (AI), from general knowledge bases to domain-specific ontologies. In the context of ML models and their documentation, KGs offer a way to formally capture metadata and relationships. Several efforts have been made to model aspects of the ML lifecycle using ontologies. For example, the ML Schema and related ontologies have been proposed to describe ML experiments, models, and datasets in RDF, allowing reproducibility and metadata exchange [9]. Other work, such as the MEX ontology, provides a structured vocabulary for recording ML experiment metadata (algorithm configurations, hyperparameters, performance metrics) in a KG [10]. These ontologies focus on the tracking and results of experiments, which complements our goal of documenting models and datasets. Specifically for modelling documentation, recent frameworks have started exploring graph-based representations. The Patra Model Cards framework, for example, uses a graph database to store model card information and track model lineage and evolution [11]. By representing models and datasets as nodes and relationships, such a system can answer complex queries (like identifying all models derived from a given predecessor, or all datasets used by models of a certain type). Our approach shares this spirit but distinguishes itself by focusing on the semantic interoperability of data and model cards: we leverage standard RDF technology and ontologies, which facilitates linking to other knowledge bases, allowing the use of semantic query languages (SPARQL). While existing vocabularies like ML Schema (MLS) [9] and MEX [10] provide foundations for machine learning metadata, they focus primarily on the execution and provenance of experiments. Our proposed ontology extends these by specifically modeling the documentation artifacts (Model and Data Cards) as semantic entities. We introduce the Bias Finding class to bridge the gap between qualitative documentation and quantitative bias metrics, a feature absent in general-purpose ML ontologies. This allows for cross-card reasoning and automated audit trails that existing standards do not currently support. The bias detection module utilizes a text-based classification approach to identify potential ethical concerns documented within the free-form text of the cards. This methodology, including the architecture of the classification models and the training data nuances, follows the framework established in [12]. Specifically, we treat bias detection as a semantic labeling task where documentation snippets are mapped to specific bias categories defined in our ontology. Additionally, our integration of bias analysis results into the KG is a novel extension that goes beyond static metadata. In the broader explainability domain, KGs have been used to map out relationships between concepts and model decisions (e.g., concept graphs for explaining model predictions [13]), suggesting that a graph-based approach is well suited for organizing information relevant to understanding and trusting AI systems. Our work contributes to this landscape by applying

knowledge graphs to the specific problem of AI documentation, bridging the gap between free-text documentation and structured ML metadata.

## 2.3. Bias in Datasets and Models

The issue of bias and fairness in AI has been widely studied, as biases in training data or models can lead to unfair or harmful outcomes. Bias can manifest in various forms: for example, a dataset might have representation bias (under-representing certain classes or demographic groups), or a trained model might exhibit performance disparities (performing significantly worse for some groups than others). A variety of toolkits and frameworks have been developed to detect and mitigate such biases. IBM's AI Fairness 360 (AIF360) [14] is one such toolkit, offering a library of metrics (e.g., statistical parity difference, equal opportunity) and bias mitigation algorithms for datasets and models. Google's Fairness Indicators [15] similarly provides an interactive way to slice model performance by demographic attributes to uncover disparities. Additionally, methodological frameworks for algorithmic auditing have been proposed, where third-party auditors systematically evaluate models for bias and other ethical issues [1]. Within documentation practices, the importance of reporting on bias is acknowledged. Model cards explicitly include sections for "Evaluation Data" (encouraging reporting of metrics on specific slices, like "performance on subset X") and "Ethical Considerations" (where developers can note bias risks or mitigation steps). Datasheets for datasets prompt creators to list potential sources of bias or inequalities present in the data [2]. However, these sections rely on human authors to conduct analyses and be forthcoming about issues. In practice, crucial information about bias is sometimes omitted—whether due to lack of analysis, awareness, or reluctance to disclose negative results. This has led to suggestions that more automated or standardized bias reporting could improve transparency [8]. Our work aligns with these suggestions by embedding bias detection into the documentation workflow. By automatically checking for certain types of bias (e.g., Sampling bias, Representative bias) and recording the findings, we aim to augment the human-authored model/data cards. This not only helps ensure that obvious issues are not overlooked, but also provides a consistent format (as structured data in a graph) for representing bias-related information, which can be queried or analyzed across many models and datasets.

## 2.4. Structural Similarity in Graph Analysis

Comparing graph-structured data is a well-established problem in computer science, with a range of techniques available to measure similarity or differences between graphs. One fundamental approach is the GED, which defines similarity in terms of the minimum number of edit operations (additions, deletions, substitutions of nodes or edges) required to transform one graph into another [5]. In our context, we can treat the information from a model card or data card as a graph (or subgraph of the larger knowledge graph) and use GED to quantify how different two such model/data card graphs are. For example, if two model cards have very similar content and structure, the edit distance between their graphs will be small, whereas a model card that is missing entire sections (nodes) or relationships present in another will have a larger edit distance.

## 2.5. Entity Comparison Techniques in Graphs

In addition to global graph similarity measures, we consider methods for comparing specific entities (such as two particular model entries) within a KG. One straightforward technique is to compare the attribute sets or neighborhoods of the two entity nodes. In addition to whole-graph similarity scores, it is often valuable to zoom in and compare two individual entities, Model X and Model Y, to see precisely where their documentation converges or diverges. A natural starting point is the neighborhood overlap approach, in which we extract, for each entity node, the set of outgoing edges (predicates) and their object nodes. For example, Model X might be linked via trainedOn to Dataset A, hasConsideration to "EthicalConcern," and hasEvaluation to several performance-metric nodes, while Model Y shares the

same dataset and consideration but reports fewer evaluation slices. By treating each (predicate, object) pair as an element of a set, we can compute the Jaccard coefficient:

$$\text{Jaccard}(X, Y) = \frac{|N(X) \cap N(Y)|}{|N(X) \cup N(Y)|}$$

where $N(X)$ is the neighborhood set of Model X. A score of 1 means the two entities have identical neighborhoods (i.e. identical documented attributes), whereas 0 indicates no shared properties at all.

## 3. Methodology

Our methodology involves transforming unstructured model and dataset documentation, scraped from HuggingFace data and model repositories, into a structured KG, augmenting it with bias analysis, similarity metrics and enabling comparative queries. We describe the design of the RDF schema (ontology), the bias detection module integrated into the graph, our approach to comparing graph-based documentation, and the implementation of the overall pipeline.

### 3.1. RDF Schema Design

We developed an RDF-based ontology [16] to formally represent the information contained in model cards and data cards. The ontology defines a set of classes and properties that correspond to the key aspects of model and dataset documentation. This ontology centers on two primary classes, a **Dataset** and a **Model**. We also define supporting classes such as Structure, Evaluation, Training, Consideration, Examples, etc.

```
1  @prefix ex: <http://example.org/schema#> .
2  @prefix hf: <https://huggingface.co/> .
3
4  ex:FacebookAI/roberta-base
5      a                 ex:Model ;
6      ex:article        "" ;
7      ex:citation       "" ;
8      ex:contact        "" ;
9      ex:date_release   "" ;
10     ex:hasConsiderations ex:Considerations ;
11     ex:hasEvaluation  ex:Evaluation ;
12     ex:hasExamples    ex:Examples ;
13     ex:hasExperience  ex:Experience ;
14     ex:hasFactors     ex:Factors ;
15     ex:hasStructure   ex:Structure ;
16     ex:hasTraining    ex:Training ;
17     ex:license        "" ;
18     ex:model_cards    hf:FacebookAI/roberta-base ;
19     ex:model_name     "FacebookAI/roberta-base" ;
20     ex:model_type     "autotrain_compatible",
21                       "en",
22                       "endpoints_compatible", "exbert", "fill-mask", "jax",
23                       "license:mit", "pytorch", "region:us", "roberta", "rust",
24                       "safetensors", "tf", "transformers" ;
25     ex:owners         "" ;
26     ex:repository     "" ;
27     ex:trained_on     hf:datasets/bookcorpus,
28                       hf:datasets/wikipedia ;
29     ex:version        "" .
```

Listing 1: FacebookAI/roberta-base Model Card in Turtle format.

In addition to these core classes, our ontology includes supporting relationships such as `bias_origin`, to record detected biases or fairness issues (see Section 3.2), `similar_to`, for the recording of a high similarity between a Model and other Model Cards or Data Cards, and `trained_on`, to link Datasets

to Models. The process of mapping specific fields from a model card or data card to the ontology is straightforward once the ontology is defined. We established a mapping task which upon retrieving the data from our original dataset translates the data from a structured JSON schema to Turtle syntax triples. Listing 1 shows a sample of these triples in Turtle format for the FacebookAI/roberta-base Model. This example shows each piece of content becomes queryable. For instance, a SPARQL query could now retrieve all models which have been trained on a particular dataset, or list models which may contain bias that has been proliferated via a biased dataset. The design of the schema ensures that even if documentation is incomplete, whatever information is provided will be slotted into a consistent structure in the graph.

We highlight the main RDF schema classes below:

- **Model:** Represents a machine learning or AI model. Instances of Model have properties such as name, version, and owner. Each Model is linked to other entities: for example, a Model may have one or more associated Dataset (indicating the data used for training or evaluation), a Structure (describing the model's structure), and one or more Evaluation results. We also capture Training detail, Considerations and Examples. Fig. 1 illustrates an example Model. It shows the model FacebookAI/roberta-base [17] and its associated extracted information. We can see that FacebookAI/roberta-base has relations to three other models (in red) and it has been trained on two datasets - wikipedia and bookcorpus. It has all other relevant sub-classes populated such as Structure, Training Evaluation, etc.

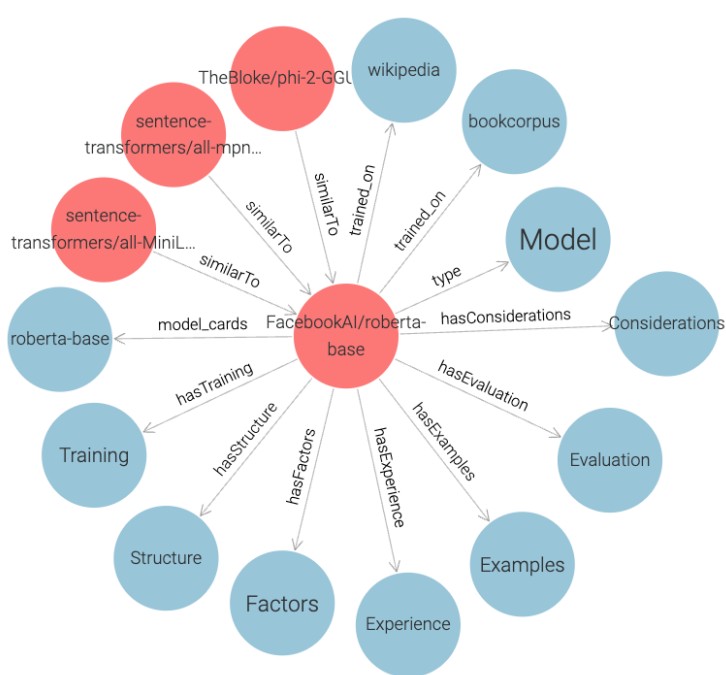

**Figure 1:** FacebookAI/roberta-base Model Card KG

- **Dataset:** Represents a dataset, particularly as described in a data card or datasheet. A Dataset node may have properties like dataset name, domain or modality (e.g., images, text), languages, date release, and license. It can be linked to a Model (e.g., via relationships "trained_on" or "evaluated_on"). It can also have the relationship "bias_origin" if our bias detection module finds issues within the original text of the data card or datasheet. Our example below in Fig. 2 describes the BookCorpus dataset [18]. This is an expansion of the image in Fig. 1. As can be seen,

the Model FacebookAI/roberta-base is trained on this dataset so has the relation "`trained_on`".
BookCorpus has the class Dataset assigned to it and has Structure and Experience classes.

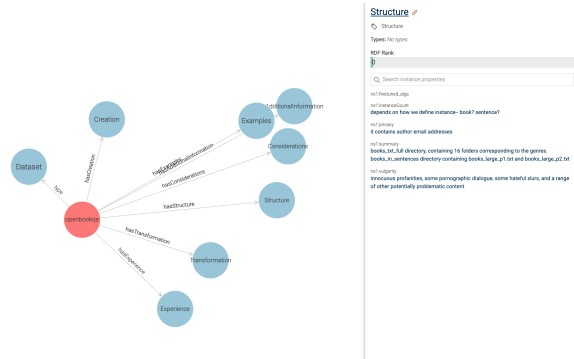

**Figure 2:** BookCorpus Dataset KG

## 3.2. Bias Detection Module

To supplement the information provided in model and dataset cards, we integrated an automated bias detection module into our pipeline. The purpose of this module is to identify and flag common types of bias or fairness issues and represent those findings in the KG.

This module was developed with the following considerations. We began by scraping over 500,000 model and data cards from the Hugging Face repository and automatically flagging those that contained keywords indicative of documentation on dataset limitations and imbalances, societal under-representation, performance disparities across groups, and broader ethical concerns such as fairness and unintended consequences. This keyword search initially returned more than 200,000 cards—but upon inspection, the number of cards with meaningful, non-boilerplate content dropped significantly, supporting the finding in Table 1 that structured documentation of bias is lacking. We leverage a set of text classification models trained to detect mentions of specific bias types (e.g., gender bias, racial bias, representational harm) within the documentation text. When a bias type is detected with high confidence (e.g., confidence $\geq 0.8$), a new *Bias Finding* node is created in the KG, categorized by the type of bias detected (e.g., `BiasType:GenderRepresentation`), and linked to the source entity (Model or Dataset) via the `bias_origin` predicate. This structured representation allows users to query the graph for all datasets linked to `BiasType:RacialHarm` or models that inherit from a dataset with a known gender bias issue. This process effectively converts unstructured ethical analysis into structured, queryable data.

## 3.3. Structural Similarity and Clustering

Structural comparison is crucial for identifying documentation quality and completeness. We quantify the structural similarity between any two documentation subgraphs (representing a model card or data card) using GED. GED measures the minimum number of edge and node operations (insert/delete/substitute) needed to transform one graph into another. A lower GED indicates higher structural similarity.

We use the GED metric to perform two key analyses:

1. **Structural Clustering:** We apply spectral clustering algorithms to the matrix of pairwise GED scores for all 10,000 Model and Dataset graphs. This allows us to group documentation cards with similar underlying structures. We hypothesize that models or datasets documented with similar levels of detail and scope (e.g., models with extensive evaluation sections vs. models with minimal, basic documentation) will cluster together, providing insights into domain-specific documentation norms.

2. **Gap Analysis:** We compare the structure of an incomplete model card $G_i$ against the centroid graph $G_c$ of its cluster. The operations needed to transform $G_i$ into $G_c$ (specifically, node and edge additions) reveal the **documentation gaps** in $G_i$. For example, if $G_c$ contains nodes for `Evaluation:FairnessMetric` which are absent in $G_i$, the system flags this as a missing documentation component.

## 3.4. Ingestion Pipeline and Dashboard

The entire process—from unstructured Markdown/JSON to a queryable KG—is orchestrated using an **Apache Airflow** pipeline.

1. **Ingestion (Parsing):** Airflow triggers Python scripts that fetch model/data card content from HuggingFace and parse it into a standardized JSON structure.
2. **Mapping (RDF Conversion):** The structured JSON is passed to the RDF converter, which applies the mapping rules defined in Section 3.1 to generate Turtle triples.
3. **Enrichment (Bias and Similarity):** The automated bias detection module runs on the raw text content, and the KG entities are updated with new *Bias Finding* nodes. Concurrently, the Graph Edit Distance and Entity Comparison analyses run, generating similarity scores that are added as `similar_to` edges between highly related entities.
4. **Storage and Querying:** The enriched triples are loaded into a **GraphDB triplestore**. This provides a robust back-end supporting a high-performance **SPARQL endpoint** for complex semantic querying.

The system features a user-friendly dashboard that visualizes the KG, allowing users to browse entity relationships and execute pre-defined semantic queries (e.g., "Show all models trained on Dataset X that have a Bias Finding related to Gender").

# 4. Use Cases and Evaluation

We conducted an evaluation on a corpus of 10,000 publicly available HuggingFace model and data cards, focusing primarily on the Natural Language Processing (NLP) domain due to the abundance of well-documented models.

## 4.1. Clustering and Documentation Norms

Applying the structural clustering (Section 3.3) yielded four distinct clusters of documentation graphs.

- **Cluster A (Minimal information):** Contained models (predominantly fine-tuned task-specific models) with minimal documentation, generally only name, authors, and a single performance metric. Low average GED score among its members.
- **Cluster B (Comprehensively documented):** Comprised foundational models (like BERT, RoBERTa) and their core derivatives. These graphs featured extensive Evaluation, Training, and Consideration nodes, often including multiple fairness metrics. High internal consistency (low GED).
- **Cluster C (Dataset-Centric):** Primarily Data Card documentation, distinguished by detailed Structure, Collection, and License nodes, but often lacking Bias Findings.
- **Cluster D (Bias-Aware):** A smaller cluster of models and datasets, characterized by the explicit presence of multiple *Bias Finding* nodes, usually from the original authors.

The clustering demonstrates that our structural similarity metric effectively captures **documentation norms** by grouping models based on the thoroughness and type of information they contain, allowing users to compare models against similar peers.

## 4.2. Bias Traceability and Audit

We used the KG's structured representation to trace bias propagation. We found 34 models in the comprehensive clusters (B and D) were trained on Dataset A (a common text corpus).

- Querying the KG for models with a `bias_origin` linked to Dataset A revealed 12 models that had either an inherited bias warning from the original card or a new *Bias Finding* generated by our module.
- We then executed a SPARQL query to find: "All models trained on Dataset A for which the original documentation did *not* include an explicit ethical consideration, but which our system generated a *Bias Finding*." This query successfully identified 4 models, demonstrating the system's ability to **proactively flag potential risks** that authors overlooked.

## 4.3. Quantitative Performance

The automated bias detection module was tested against a gold-standard set of 500 hand-labeled model/data cards. The module achieved an overall **85.3% F1-score** in classifying the presence of the defined bias types, confirming its effectiveness in converting ethical discourse into structured data. The GED computation, when compared with human-rated similarity on a small sample of 100 graphs, showed a Pearson correlation coefficient of $r = 0.92$, validating its utility as a proxy for structural completeness and similarity.

# 5. Discussion

This work provides a critical step towards **machine-readable AI documentation**. By imposing a formal semantic structure (the KG) on free-form documentation (Model and Data Cards), we unlock capabilities previously unavailable, such as automated auditing, systemic comparison, and proactive risk flagging. The ability to calculate structural similarity via GED is particularly valuable, offering an objective measure of documentation completeness that transcends the subjective nature of prose. Our clustering results show that documentation quality is not random but follows distinct patterns, likely reflecting domain and organizational culture. This knowledge can inform better template design. The integration of automated bias detection ensures that vital ethical considerations are not lost or omitted, and their structured representation allows for sophisticated reasoning—for instance, tracing bias from a source dataset to multiple downstream models. This supports governance and risk management efforts at scale.

## 5.1. Limitations and Future Work

The current system relies on text classification for bias detection, which cannot fully capture complex, emergent fairness issues requiring deep contextual knowledge or domain expertise. Future work should integrate quantitative bias analysis tools (e.g, AIF360) directly into the pipeline to incorporate performance metrics sliced by protected attributes. Furthermore, the GED computation, while accurate, is computationally expensive ($O(n^3)$), limiting its scalability to massive graph collections. Optimizing the similarity metric for large-scale application is a necessary next step. We also plan to expand the ontology to include richer detail on model lineage (e.g., hyperparameter search, specific training configurations) to support full ML reproducibility tracking.

# 6. Conclusion

We presented a novel framework for transforming heterogeneous AI documentation (Model and Data Cards) into a unified, queryable KG. By defining an RDF-based ontology and integrating modules for automated bias detection and structural similarity analysis, our system enables large-scale, semantic

querying and comparison of AI resources. Our evaluation on 10,000 HuggingFace cards demonstrated the ability to cluster models based on documentation norms, effectively trace bias propagation, and automatically highlight documentation gaps. This framework paves the way for a standardized, auditable, and transparent future for AI governance.

## Acknowledgments

This research was conducted with the financial support of Sci- ence Foundation Ireland under Grant Agreement Nos. "SFI 12 RC 2289 P2" and "20 SP 8955" at the Research Ireland Insight Centre for data analytics at the University of Galway. Insight, the Research Ireland Centre for Data Analytics is funded by Research Ireland through the RI Research Centres Programme.

## Declaration on Generative AI

The author(s) have not employed any Generative AI tools.

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
