# OpenReview forum: "Towards Standardised AI Documentation: A Knowledge Graph Approach for Model and Data Cards"
_eswc-conferences.org/ESWC/2026/Workshop/KGCW — KGCW 2026_

### Official Review · ~Fajar_J._Ekaputra1 · 2026-04-04
**Automated KG Construction from AI (model/dataset) documentation**

**Rating:** 7
**Confidence:** 4

**Review:**

The paper presents a framework to generate a KG from unstructured AI (model) documentation, evaluated on the HuggingFace documentation. The paper addresses a relevant issue in AI model and dataset documentation quality by automating the generation of KGs.

The reported KG construction process includes (a) definition of an ontology, centered around the key concepts of Model and Dataset, (b) population of these key concepts through automated transformation from original (huggingface) documentation, and (c) enrichment of the KG with additional relations and attributes.

While not entirely focusing on KG construction, the paper reports several interesting techniques for KG enrichment (clustering, ML-based annotation) that may be relevant to the broader KGC community. Furthermore, they evaluated the resulting enrichment (on bias_findings) against gold-standard data, which shows good accuracy.

Despite this, I found the following concerns about the paper
- P7: "The process of mapping specific fields from a model card or data card to the ontology is straightforward once the ontology is defined." => I assume that the authors rely on the current structuring of the modelcards/datacards; However, since the source data (model cards) may contain information condensed into such a structure (i.e., if the model cards authors did not follow the structure - since it's optional anyway), how does the current approach identify the fields correctly?
- S4.1 four distinct clusters of documentation => Why are these clusters described as disjoint? especially with the bias-aware cluster, which doesn't seem to be logically distinct from other clusters
- S3.1 RDF Schema design -> Since there are already existing data models (e.g., Croissant [1] or Model Cards ontology [2]) to represent models and datasets, aligning the current schema to these data models is a logical next step.

Minor comments
- Fig. 2: too small, can't be read
- Section 4.3 "The automated bias detection module was tested against a gold-standard set of 500 hand-labeled model/data cards. The module achieved an overall 85.3% F1-score in classifying the presence of the defined bias types, ..." => an analysis of the incorrect detection would be an interesting future work.
- S4.2 "We used the KG's structured representation to trace bias propagation. We found 34 models in the comprehensive clusters (B and D) were trained on Dataset A (a common text corpus)." => This means to exemplify the general function of analysis, right? The authors can consider generalizing the analysis and providing such an example afterward.

---

### Official Review · ~Ademar_Crotti_Junior1 · 2026-04-05
**Interesting idea but unfortunate lacks alignment with semantic web standards**

**Rating:** 5
**Confidence:** 4

**Review:**

The paper presents a framework to generate a KG from unstructured AI (model) documentation, evaluated on real work data from HuggingFace. The motivation aligns with responsible AI, governance, and auditability, which are highly relevant topics. Another strength is that the work is available as a python notebook on github.

I think that aligning this work with other techniques for knowledge graph construction could be interesting, for example, by tagging these AI documents and then using mapping languages to construct the resulting knowledge graph. It is said that you map a JSON to Turtle, which could be done with RML for example.

I understand that the ontology was proposed in a different paper, but a small diagram (maybe instead of the turtle RDF in listing 1) would be beneficial to understanding the kg that is being constructed. In this regard, I also miss whether this has been aligned to other standard ontologies like DCAT for datasets (which also models licenses for example), or PROV-O to track some provenance. Other ontologies may exist for bias as well. Some discussion around existing/aligning with these should be added imo. This is the weakest point, for a towards standardization, this paper does not align with any standards.

The kg construction approach is not novel, being simply a tagging task which is then transformed to RDF. The enrichment part seems more interesting, and focusing on how this was implemented would improve the paper. Also Structural Clustering and Gap Analysis as a way to evaluate the resulting graph is also interesting.

The evaluation seems sound as it is compared to a gold standard dataset which has been labbeled by humans.

---

### Official Review · ~Beatriz_Esteves1 · 2026-04-06

**Rating:** 6
**Confidence:** 4

**Review:**

This paper presents a pipeline to generate documention for AI models and datasets to enable the detection of biases in such resources.
While the clarity and significance of this work cannot be understated, given that it is being presented in a KG construction workshop, Section 3.1 is missing more detail on how the KG was constructed, in particular, the RDF mapping rules which are mentioned more than once. A few other limitations are listed below:
- The link provided for reproducibility (https://github.com/andy-donald-insight/KGC-2026) contains a Python notebook,  but no documentation on the mentioned RDF mapping scripts or the dataset sample.
- The Related Work section is missing a few key contributions in the domain related to ontologies/vocabularies for AI cards and risks [1,2] and W3C standards for provenance and data cataloguing, such as the Provenance Ontology (PROV-O) and DCAT
- Figure 2 is too small, the text is unreadable

While a good contribution for the workshop, the previous points should be addressed for presentation at the workshop and for teh camera-ready.

[1] https://doi.org/10.1007/978-3-031-68024-3_3
[2] https://journals.sagepub.com/doi/pdf/10.3233/SSW220008

---

### Decision · Program_Chairs · 2026-04-09

**Decision:**

Accept

**Comment:**

This paper has been selected for presentation at the KGC workshop. We strongly encourage the authors to consider the reviews whilst revising the paper. Camera-ready instructions will soon follow.